# Benchmarking AlphaMissense pathogenicity predictions against cystic fibrosis variants

Eli Fritz McDonald [1,2], Kathryn E. Oliver [3,4], Jonathan P. Schlebach[5], Jens Meiler [1,2,6,7]*, Lars Plate [1,8,9]*

1 Department of Chemistry, Vanderbilt University, Nashville, Tennessee, United States of America, 2 Center for Structural Biology, Vanderbilt University, Nashville, Tennessee, United States of America, 3 Department of Pediatrics, Emory University School of Medicine, Atlanta, Georgia, United States of America, 4 Center for Cystic Fibrosis and Airways Diseases, Children's Healthcare of Atlanta and Emory University, Atlanta, Georgia, United States of America, 5 Department of Chemistry, Purdue University, West Lafyette, Indiana, United States of America, 6 Department of Pharmacology, Vanderbilt University, Nashville, Tennessee, United States of America, 7 Institute for Drug Discovery, Leipzig University, Leipzig, Germany, 8 Department of Biological Sciences, Vanderbilt University, Nashville, Tennessee, United States of America, 9 Department of Pathology, Microbiology and Immunology, Vanderbilt University Medical Center, Nashville, Tennessee, United States of America

* jens@meilerlab.org (JM); lars.plate@vanderbilt.edu (LP)

**Data Availability Statement:** All relevant data are within the manuscript and its Supporting information files.

## Abstract

Variants in the cystic fibrosis transmembrane conductance regulator gene (*CFTR*) result in cystic fibrosis–a lethal autosomal recessive disorder. Missense variants that alter a single amino acid in the CFTR protein are among the most common cystic fibrosis variants, yet tools for accurately predicting molecular consequences of missense variants have been limited to date. AlphaMissense (AM) is a new technology that predicts the pathogenicity of missense variants based on dual learned protein structure and evolutionary features. Here, we evaluated the ability of AM to predict the pathogenicity of CFTR missense variants. AM predicted a high pathogenicity for CFTR residues overall, resulting in a high false positive rate and fair classification performance on CF variants from the CFTR2.org database. AM pathogenicity score correlated modestly with pathogenicity metrics from persons with CF including sweat chloride level, pancreatic insufficiency rate, and *Pseudomonas aeruginosa* infection rate. Correlation was also modest with CFTR trafficking and folding competency *in vitro*. By contrast, the AM score correlated well with CFTR channel function *in vitro*–demonstrating the dual structure and evolutionary training approach learns important functional information despite lacking such data during training. Different performance across metrics indicated AM may determine if polymorphisms in CFTR are recessive CF variants yet cannot differentiate mechanistic effects or the nature of pathophysiology. Finally, AM predictions offered limited utility to inform on the pharmacological response of CF variants i.e., *theratype*. Development of new approaches to differentiate the biochemical and pharmacological properties of CFTR variants is therefore still needed to refine the targeting of emerging precision CF therapeutics.

**Funding:** This work was supported by R35
GM133552 (NIGMS), R01 HL167046 (NHBLI),
R00HL151965 (NIH) and OLIVER22A0-KB (CFF).
EFM was supported by a predoctoral fellowship
F31 HL162483 (NHLBI) and Chemical-Biology
Interface training grant T32 GM065086 (NIGMS).

**Competing interests:** The authors have declared
that no competing interests exist.

## Introduction

Cystic fibrosis (CF) is a lethal genetic disease caused by variants in the epithelial anion channel cystic fibrosis transmembrane conductance regulator (CFTR) [1]. CFTR is composed of an N-terminal lasso motif, two nucleotide binding domains (NBDs), two transmembrane domains (TMDs) and an unstructured regulatory domain (RD) [2]. Loss of CFTR protein production or function results in osmotic dysregulation at the epithelium of the skin, pancreatic duct, and lungs–leading to high sweat chloride levels, pancreatic insufficiency, and lung infections respectively [3]. Standard treatment paradigms for CF involve supplementation of salt, vitamins, and digestive enzymes, together with airway clearance therapies and small molecule CFTR modulators known as *potentiators* and *correctors*. CFTR variants experience distinct structural defects and proteostasis states leading to divergent pharmacological response profiles to modulators also known as *theratypes* [4–7].

At present, elexacaftor-tezacaftor-ivacaftor (ETI) is the best available highly effective modulator therapy for CF. This triple combination is clinically approved for ~170 CFTR variants, including the most commonly reported allele, deletion of phenylalanine 508 (F508del) [8–11]. ETI is composed of one gating potentiator (ivacaftor, VX-770) and two protein maturation correctors, tezacaftor (VX-661) and elexacaftor (VX-445). The corrector compounds have been suggested to directly bind unique subdomains of CFTR: VX-661 to TMD1 [12, 13], and VX-445 to the N-terminal lasso and TMD2 [14, 15]. Correctors contribute intermolecular interactions that favor the properly folded, trafficking competent state of CFTR. Due to the distinct binding sites, VX-661 and VX-445 elicit different mechanisms of action and confer variable theratype responses. Thus, profiling CFTR variant theratypes to these and other emerging modulators remains an important priority for CF personalized medicine.

Increasing implementation of next-generation sequencing approaches for CFTR DNA analysis has rapidly augmented the pace of novel CFTR variant discovery; and thus, hastened the need for more accurate pathogenicity prediction tools. This is particularly relevant to individuals with CFTR related metabolic syndrome (CRMS), also known as CF Screen Positive Inconclusive Diagnosis (CFSPID). Patients are diagnosed with this condition if they possess a positive newborn screen for CF and either of the following criteria: (1) normal sweat chloride value ($<$30 mEq/L) and two identified CFTR variants, at least one of which exhibits unclear phenotypic consequences; or (2) intermediate sweat chloride value (30–59 mEq/L) and detection of one or zero CF-causing variants [16]. Clinical symptoms worsen for approximately 11–48% of CRMS/CFSPID patients, who eventually convert to a CF diagnosis [17, 18]. Insufficient data exists to predict which CFTR variants (or other factors) enhance the risk for progression to CF.

Furthermore, high-throughput methods for characterizing CFTR variant severity are limited. Only 804 of the reported 2,111 variants have been annotated for disease association according to *in vitro* or clinical data [19]. The majority of these CFTR variants are single amino acid substitutions or missense variants [20]. Recently, AlphaMissense (AM) was published as a technology designed to predict the pathogenicity of missense variants throughout the human proteome [21]. Among well-characterized genetic diseases, AM included CF pathogenicity predictions for every possible CFTR single amino acid substitution. AM provides a significant advance beyond previous attempts to model a limited number of CFTR variants [22, 23]. Here, we evaluated the predictive validity of AM across several metrics of CF data such as pathogenicity in people with CF, *in vitro* CFTR folding and function, and theratype.

The increasing pace of novel CFTR variant discovery has created a need for pathogenicity prediction, especially among wild-type (WT) heterozygous individuals, e.g. carriers, and CRMS/CFSPID individuals whose variants remain uncharacterized. Our analysis suggests AM

predicts the relative pathogenicity of severe CF-causing variants well, while performing modestly for variants of unknown significance (VUS) or variants of varying clinical consequence (VVCC). Overall, AM showed a high false positive rate for predicting CFTR2 patient outcomes [19]. Among VUSs, two variants from CFTR2 and 368 variants from ClinVar [24] were predicted as pathogenic. By contrast, the S912L VUS from CFTR2 was predicted benign despite clinical outcomes indicating ~half the people with this variant display hallmarks of CF disease. AM scores correlated modestly with CF pathogenicity metrics and CFTR trafficking/folding competency *in vitro*. Correlation improved when compared to CFTR channel functional data. These analyses imply AM has learned important trends in variant function despite not training on such data. Finally, we provide evidence that AM offers little power in predicting CFTR variant theratype, although we note this measure is beyond its intended design. Thus, AM may offer capabilities in predicting the pathogenicity of emerging variants but proved less useful for theratyping variants.

## Results and discussion

### I. AlphaMissense predictions of CFTR pathogenicity

AM makes pathogenicity predictions based on a 90% accuracy against ClinVar data [21]. For CFTR, AM predicted scores from 0.56–1.00 as pathogenic, scores from 0.34–0.56 as ambiguous, and scores from 0.04–0.34 as benign. We mapped the average AM prediction score per residue onto a CFTR structure (PDBID 5UAK) [25] (Fig 1A). TMDs showed a propensity for pathogenicity in contrast to residue conservation as calculated by ConSurf [26], which suggested the TMDs are comparatively variable across species (Fig 1A & S1A Fig). Since the regulatory domain (RD) is disordered and not resolved in the CFTR structure 5UAK, we also plotted the average AM score for RD residues against a RD map highlighting key features such as transiently formed α-helices and phosphorylation sites [4] (Fig 1B). Despite noted difficulty with disordered regions [21], AM predicted RD residues ~760–775 as a hotspot for pathogenicity. This is consistent with the role of transient helix 752–778 in CFTR gating through interactions with a conserved region of the NBD2 C-terminus [27, 28].

We sought to evaluate AM's ability to correctly predict CF pathogenesis on 169 classified missense variants from CFTR2.org [19]. The CFTR2 database offered a rich patient metric repository including pathogenicity classifications as CF-causing, VVCC, non-CF-causing, or VUS (S1 Table). VVCC are defined by CFTR2 as variants that may cause CF when found heterozygous with CF-causing variants, which results in variable clinical diagnosis of CF, e.g., a person with a VVCC and a CF-causing variant may or may not present with CF [19].

AM showed a 95% accuracy (104/110) for predicting pathogenic variants and a 78% accuracy (14/18) for predicting benign variants based on variant determination in CFTR2 (S1 Table). We calculated the receiver operating characteristic (ROC) curve for all pairwise comparisons of pathogenicity predicted by AM (Fig 1C, See Methods). Briefly, all pairwise comparisons were considered–pathogenic, ambiguous, or benign were taken in turn to be a true positive. The alternative two predictions for a specific comparison were taken to be false positives. We considered pathogenic to predict CF-causing, ambiguous to predict VVCC, and benign to predict non-CF causing, VUS were not used. While looping through all possible score thresholds, the corresponding true positive and false positive rates were calculated and plotted. Benign predictions showed the highest area under the curve (AUC) (0.91) followed by pathogenic (0.80) and ambiguous respectively (0.66)–suggesting that AM has a high false positive rate, particularly for ambiguous predictions (Fig 1C). A high false positive rate may be attributed to a poor AlphaFold2 (AF2) predicted structure of CFTR. However, the AF2

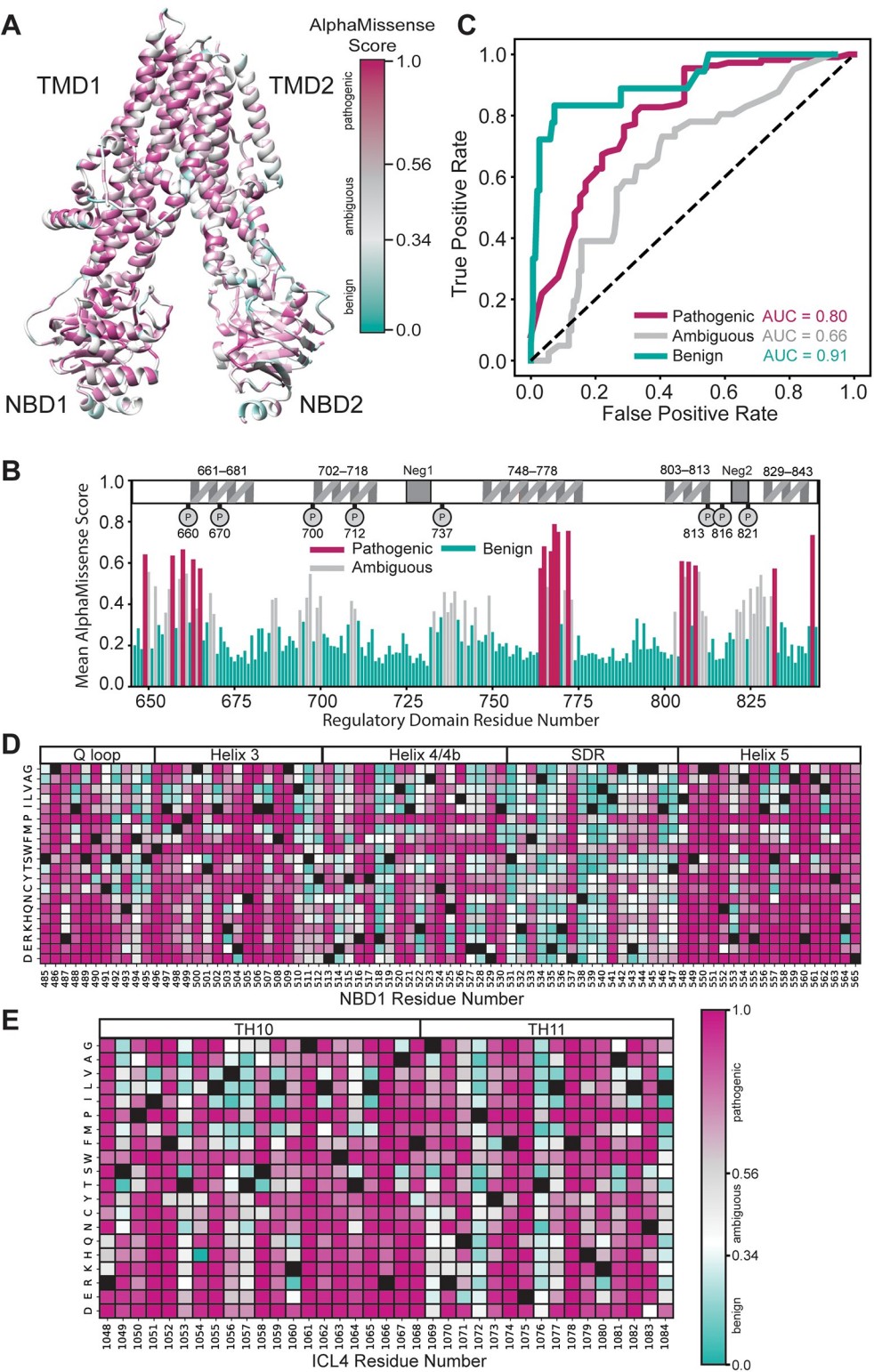

**Fig 1. AlphaMissense predictions of CFTR pathogenicity compared to CFTR2.org repository. A.** The average AM score per residue mapped onto the CFTR structure (PDBID 5uak) [4]. Variants with a score from 0.56–1.00 were classified by AM as pathogenic, variants with a score from 0.34–0.56 were classified as ambiguous, and variants with a score from 0.04–0.34 were classified as benign. **B.** The average AM score for the regulatory domain (RD) with an RD map of important features for reference [4]. Transient helices shown, two negatively charged regions (Neg1/2), and

important phosphorylation sites shown as P. Notably, the second half of the transient helix 748–778 is predicted to be a hotspot of RD variant pathogenicity. **C**. Receiver operating characteristic (ROC) curve of AM predictions benchmarked against 169 CFTR2.org database classifications. Our curated data set contained 110 CF-causing, 41 variants variable clinical consequence (VVCC), and 18 non-CF causing missense variants. In the pathogenic curve (violet)–pathogenic prediction of a CF-causing variant was considered a true positive. Likewise in the ambiguous curve (grey)–ambiguous prediction of a variable clinical outcomes was considered a true positive. Finally, in the benign curve (bluegreen)–benign prediction of a non-CF-causing variant was considered a true positive. **D**. Heatmap of AM scores for NBD1 residues 485–565 including all regions interfacing with ICL4 and adjacent regions. Pathogenicity colored as in **A**. and WT residues depicted in black. **E**. Heatmap of AM scores for ICL4 residues 1048–1084, colored as in **E**.

predicted CFTR [29] shows a root mean squared deviation of just 2.5 Å from the active state cryo-EM model (PDB ID 6MSM, resolution 3.2 Å [30]) (S1B Fig).

We noted seven VUS in the CFTR2.org database and their respective AM predictions (Table 1). The location of these variants in the CFTR structure is shown (S1C Fig). Benign predicted R31L disrupts the arginine framed tripeptide motif at R29-R31, important for folding evaluation prior to ER export [31] and may affect endocytosis rates [32]. V201M was ambiguously predicted, consistent with our previous report describing this variant as mildly mis-trafficked and selectively sensitive to VX-661 [23]. A439V (benign prediction) and Y1014C (ambiguous prediction) showed trafficking and function slightly below WT [33] suggesting these variants are benign. Benign predicted variant S912L lies close to the CFTR glycosylation sites at N894 and N900, thus we speculated this mutation could interfere with glycan processing. Nevertheless, S912L trafficking and function remained sufficient compared to WT *in vitro* [34].

Variants D924N and M952T, both located in transmembrane helix 8, are predicted as pathogenic (S1C Fig). D924N resides in the potentiator binding hotspot [35, 36] and, according to clinical data, may cause pancreatic insufficiency but not lung disease [37]. M952T displays robust functional expression *in vitro* [38], and two patients with an M952T/F508del genotype exhibit normal chloride transport measured from intestinal mucosa [38]–suggesting this variant is likely not pathogenic, despite the AM prediction.

For performance comparison, we also plotted ROC curves for AM predictions of the 115 ClinVar variants from the AlphaMissense study and observed 96% average accuracy as presented previously [21] (S1D Fig). To validate this finding, we additionally downloaded a dataset of 209 ClinVar variants directly from ClinVar, including 96 overlapping variants from the AlphaMissense benchmark set. The ROC curve for our expanded ClinVar dataset showed >90% prediction accuracy with additional variants (S1E Fig). Finally, we plotted a ROC curve for 113 ClinVar variants not included in the AM benchmark set, which revealed a >90%

**Table 1. CFTR2.org Variants of Unknown Significance (VUS).** Seven missense variants of unknown significance (VUS) from the CFTR2.org database with their respective AM scores and predicted pathogenicity. Variants D923N and M952T are predicted to be pathogenic.

| Variant | AlphaMissense Score | Pathogenicity Prediction |
|---------|---------------------|--------------------------|
| R31L    | 0.17                | benign                   |
| V201M   | 0.36                | ambiguous                |
| A349V   | 0.26                | benign                   |
| S912L   | 0.12                | benign                   |
| D924N   | 0.83                | pathogenic               |
| M952T   | 0.85                | pathogenic               |
| Y1014C  | 0.37                | ambiguous                |

accuracy and indicates AM performs well on ClinVar data outside of the training set (S1F and S1G Fig). In addition to classified variants used for performance evaluation, ClinVar contains 1,277 CFTR VUS [24]. AM predicted VUS ClinVar variants to contain 728 benign, 181 ambiguous, and 368 pathogenic variants (S1H Fig, S2 Table).

AM performance was also compared to two other pathogenicity prediction tools, Evolutionary Scale Modeling (ESM) [39] and Evolutionary model of Variant Effect (EVE) [40] (S2A and S2B Fig). In the ROC AUCs of benign variants, the AM value (0.91) was higher than those obtained for ESM (0.78) or EVE (0.78). A similar observation was made for pathogenic ROC AUCs, with AM (0.80) slightly above ESM (0.76) or EVE (0.73). ROC AUCs for ambiguous variants were nearly uniform across all methods (AM, 0.66; ESM, 0.65; EVE, 0.64). AM therefore offers a slight advantage for predicting pathogenic or benign variants and less utility regarding ambiguous variants.

Previous analysis of CFTR variants across sampled genetic information indicates the NBD1-intracellular loop 4 (ICL4) interface is a hotspot of pathogenicity [41]. Thus, we generated a heatmap of AM scores for NBD1 residues 485–565, which encompass the α-helical subdomain, structurally diverse region (SDR), and the entire NBD1-ICL4 boundary (Fig 1D). For the Q-loop (residues 486–495) and helix 3 (residue 496–512), potential substitutions are largely predicted as pathogenic except for residues 494 and 511. Of note, AM predicts position 508 as intolerant to substitution. Deletion of the encoded phenylalanine (F508del) is the most frequently reported variant among worldwide CF populations [8, 42–44]. Most variations calculated as benign or ambiguous occur within helix 4/4b of the α-helical subdomain (residues 511–532) or SDR (residues 533–547) (Fig 1D). Possible substitutions across helix 4/4b that are predicted as pathogenic include V520 and C524. V520F[7] and C524X [45] are not presently approved for CFTR correctors and potentiators. Most substitutions (40% benign, 35% pathogenic) within the SDR are predicted as benign as expected base on the lack of structure in the region.

In contrast, variations at the NBD1-ICL4 interface are overwhelmingly scored as severe. Residues 548–565 comprise the NBD1 core helix 5, which directly interacts with ICL4 and demonstrates the strongest sensitivity (7% benign, 82% pathogenic) to mutation with potential substitutions predicted as pathogenic (Fig 1D). This region contains numerous CF-causing variants, some of which are refractory to available CFTR modulators, such as R560T/K/S [4, 41]. Within the ICL4 region (residues 1048–1084), AM scores indicate 14% benign and 69% pathogenic predictions (Fig 1E). The heatmap reveals residues 1069, 1072, 1076, and 1084 as relatively tolerant to substitution. Together, these data suggested AM pathogenicity scores matched previous findings, as well as our general understanding about residue conservation throughout CFTR, while providing specific information about every possible substitution.

## II. Cystic Fibrosis pathogenicity correlated modestly with AlphaMissense predictions

In addition to classifying variant pathogenicity, the CFTR2.org database annotates clinical outcomes for persons with CF including sweat chloride levels, pancreatic insufficiency rates, *Pseudomonas aeruginosa* infection rates, and lung function [19]. We curated the clinical outcomes for all CFTR missense variants with available data (S1 Table, See Methods). We then analyzed the ability of AM to predict patient pathogenicity metrics. Briefly, CFTR2.org data were downloaded from the Variant List History tab and filtered for 176 missense variants (169 classified and 7 VUS). Then, clinical outcome data were manually assembled by searching each variant and recording the sweat chloride (mEq/L), pancreatic insufficiency rate (%), *P. aeruginosa*

infection rate (%), and lung function (forced expiratory volume in one second (FEV1), % predicted). Of note, CFTR2 data was based on individual alleles, e.g. missense variants.

First, we plotted AM score versus CF sweat chloride levels for 123 missense variants with sweat chloride values reported (Fig 2A). AM score correlated modestly with sweat chloride levels (Pearson Correlation Coefficient: 0.46, Spearman Correlation Coefficient: 0.48). CF-causing variants, shown in blue, clustered in the top right corner, indicative of high AM scores and elevated sweat chloride levels. By contrast, VVCCs, shown in yellow, clustered in the bottom right corner, reflecting an excessive AM score (Fig 2A). When considering CF-causing or VVCC separately, we note a reduced correlation between sweat chloride levels and AM scores (S3A and S3B Fig), suggesting AM captures the trend across all variant types rather than performing better on pathogenic variants.

Next, we plotted AM score versus pancreatic insufficiency rates for 116 missense variants present on at least one allele of persons with CF with CFTR2 outcomes reported (Fig 2B). AM scores correlated poorly with pancreatic insufficiency rates (Pearson coefficient: 0.31, Spearman Coefficient: 0.41) compared to sweat chloride. Again, AM failed to predict VVCCs, shown in yellow, on this metric (Fig 2B). However, considering CF-causing and VVCCs separately failed to change the correlation for pancreatic insufficiency (S3C and S3D Fig). Finally, we plotted AM score versus *P. aeruginosa* infection rates for 114 missense variants on at least one allele with CFTR2 outcomes reported (Fig 2C). AM correlated better here than for pancreatic insufficiency rates, but worse than for sweat chloride (Pearson Coefficient: 0.38, Spearman Coefficient: 0.44). However, it performed better on VVCCs, yet correlation was again reduced when only CF-causing or VVCCs were separately considered (S3E and S3F Fig).

Taken together, AM correlated modestly with clinical data and performed poorly on VVCCs and VUSs. For example, VUS S912L was predicted benign with an AM score of 0.12. However, this variant was associated with sweat chloride levels of 60 mEq/L (Fig 2A), which resides exactly at the diagnostic cutoff for CF. S912L displays a pancreatic insufficiency rate of 57% (Fig 2B) and *P. aeruginosa* infection rate of 50% (Fig 2C)–suggesting this variant may present with more pathologic characteristics than predicted or annotated in CFTR2. Unfortunately, pathogenic-predicted variants such as D924N and M952T have insufficient data available on CFTR2 for comparison. Weak performance by AM could be attributable to high false positive rates and/or compound heterozygous genotypes. The latter factor likely complicates interpretation of clinical data, as people with complex CF alleles may exhibit differing degrees of variant severity on each chromosome (e.g. one CF-causing paired with a VUS/VVCC) compared to patients with the same variant severity on each allele (e.g. two CF-causing).

## III. AlphaMissense predicts CFTR function beyond folding and trafficking competency

Much CFTR biochemical and functional data was also available for comparison, including recent deep mutational scanning (DMS), theratype screening, and spatial covariance studies [23, 33, 46] (S3 and S4 Tables). In the DMS study, fluorescence-activated cell sorting was used to measure the cell surface immunostaining intensity of an epitope-tagged library of 129 CFTR variants including 100 missense variants [23]. In the theratype study, 655 variants including 585 missense variants were screened for their trafficking efficiency and function [33]. In the spatial covariance study, a CFTR trafficking and a chloride conductance index were established to characterize variant temperature response [46]. Variable, albeit high, overlap was observed between the CFTR2 dataset and the *in vitro* data sets discussed below (S2C Fig).

First, we evaluated AM ability to predict CFTR folding competency–which is well characterized to correlate with cell surface expression and trafficking efficiency [47–50]. We plotted

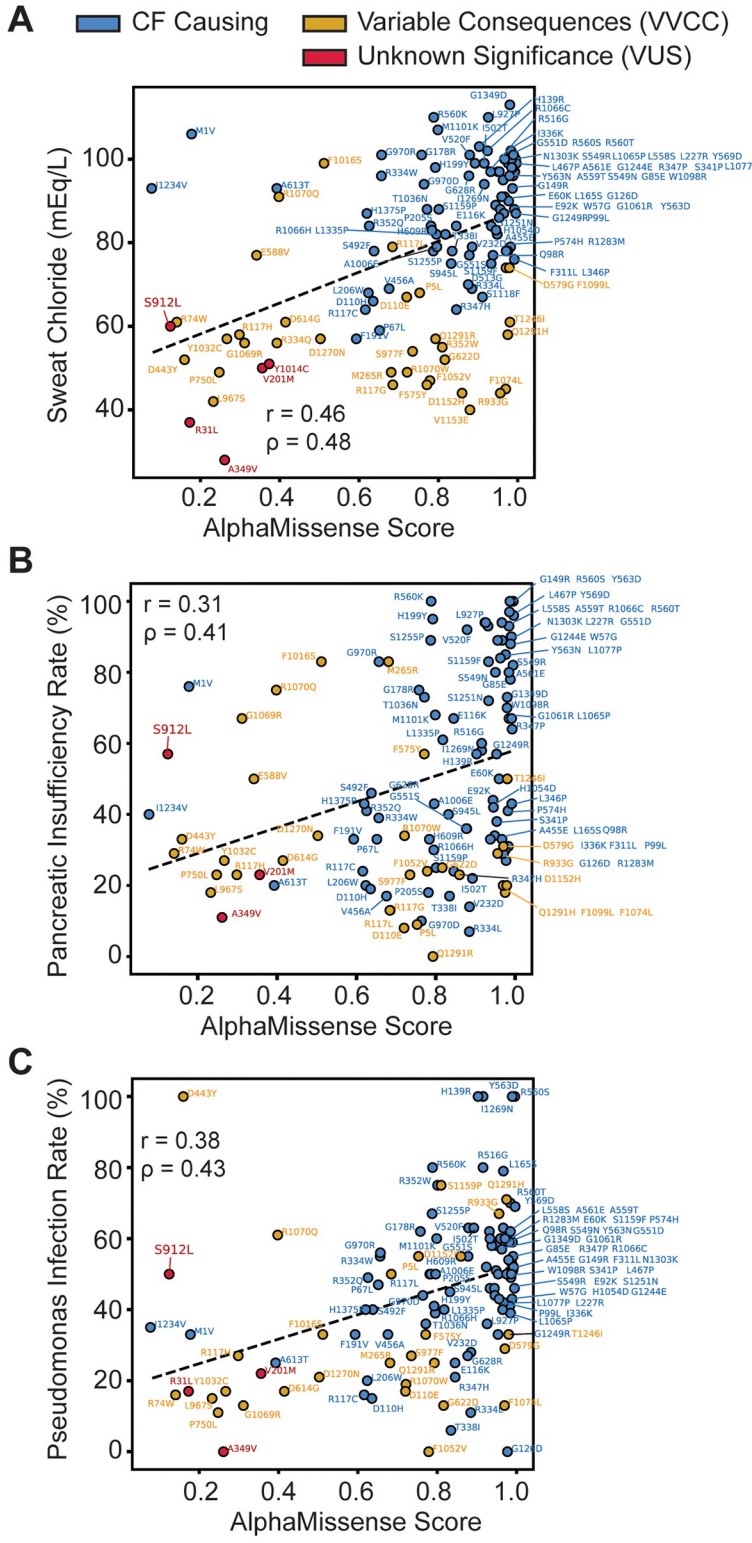

**Fig 2. Benchmarking AlphaMissense against cystic fibrosis patient pathogenicity metrics. A**. AM score plotted against sweat chloride levels in milliequivalents per liter (mEq/L) for 123 missense variants. Healthy sweat chloride levels were <30 mEq/L. CF-causing variants were shown in blue, variants of variable clinical consequence (VVCC) were shown in yellow, and variants of unknown significance (VUS) were shown in red (as annotated in CFTR2). A modest linear correlation (Pearson Coefficient r = 0.46, Spearman Coefficient ρ = 0.48) was observed. **B**. AM score

plotted against pancreatic insufficiency rates in percent for 116 missense variants. Less correlation (Pearson Coefficient r = 0.31, Spearman Coefficient ρ = 0.41) was observed than with sweat chloride, notably many VVCCs were predicted pathogenic AM score but demonstrate a low pancreatic insufficiency rate. Colors annotated as in **A. C**. AM score plotted against *P. aeruginosa* infection rates in percent for 114 missense variants. Colors annotated as in **A**. Linear correlation (Pearson Coefficient r = 0.38, Spearman Coefficient ρ = 0.43) is shown. Interestingly, S912L, a variant of unknown significance is predicted to be benign but shows a high sweat chloride, pancreatic sufficiency, and *P. aeruginosa* infection rate compared to variants with similar scores.

AM prediction scores for 100 missense variants versus DMS cell immunostaining intensity (Fig 3A, S4 Table), which showed an inverse relationship with poor correlation (Pearson coefficient: -0.37, Spearman Coefficient: -0.37). Notably, among variants in the top right corner, e.g. high AM score and high surface staining, we observed several gating variants (G551D/S, R347H, S1251N, and G1244E etc.) (Fig 3A). Mis-gating variants traffic normally, but they are CF-causing due to disrupted properties of channel opening and closing. This result demonstrated that AM failed to infer the nature of the variant defect.

Next, we plotted AM prediction scores for 538 missense variants versus CFTR trafficking efficacy as measured by the ratio of mature, fully-glycosylated CFTR (band C) to the immature glycoform (band B) on western blot (C/B band ratio) (Fig 3B, S3 Table) [33]. Experimental data was filtered for plotting clarity (S4 Fig, See Methods). We removed highly variable experimental data with a standard error of the mean (SEM) greater than 30. Most CFTR variants show a C:B ratio less than 30% of WT, indicating a lack of reproducibility for these measurements with higher variability (8% of data points removed, 92% retained). AM scores displayed improved inverse correlation with the larger trafficking efficiency dataset (Pearson coefficient: -0.50, Spearman Coefficient: -0.53). This finding suggested AM can predict CFTR folding competency across diverse types of variants. Several off-axis variants were annotated that show poor predictions and poor trafficking (<30% of WT) based on the distribution of all trafficking data (S4A and S4B Fig).

Finally, we evaluated AM ability to predict CFTR function as measured by transepithelial current clamp conductance [33]. We plotted AM prediction scores versus forskolin (FSK)-induced basal CFTR channel activity as percent WT (FSK %WT) for 546 missense variants (Fig 3C). Again, highly variable experimental data were filtered out considering an SEM greater than 20 as most variants were less than 20% of WT (S4 Fig, See Methods), leaving 93% of the experimental data for comparison to AM. AM scores inversely correlated best with CFTR function measured by conductance (Pearson coefficient: -0.70, Spearman Coefficient: -0.69). Several off-axis variants were noted which show poor predictions and poor channel function (<30% of WT) based on the distribution of functional data (S4C and S4D Fig).

We verified the increased capability to predict CFTR function by correlating AM scores with a spatial covariance study (S5 Fig). This study describes trafficking (measured by western blot band shift assay) and chloride conductance indices and presents data for both metrics at 37 ºC and reduced temperature (27 ºC) [46]. Reduced temperature is a well-established method for partially rescuing F508del biogenesis [51]. We observed a modest correlation (Pearson coefficient: -0.46, Spearman Coefficient: -0.44) with trafficking index at 37 ºC, and a similar correlation at 27 ºC (Pearson coefficient: -0.48, Spearman Coefficient: -0.49) (S5A and S5B Fig). Again, correlation increased when compared to chloride conductance index (Pearson coefficient: -0.58, Spearman Coefficient: -0.54 at 37 ºC vs. Pearson coefficient: -0.50, Spearman Coefficient: -0.53 at 27 ºC) (S5C and S5D Fig). Together these results indicated that AM scores are closely aligned with pathogenicity but cannot differentiate between variants that compromise expression versus function.

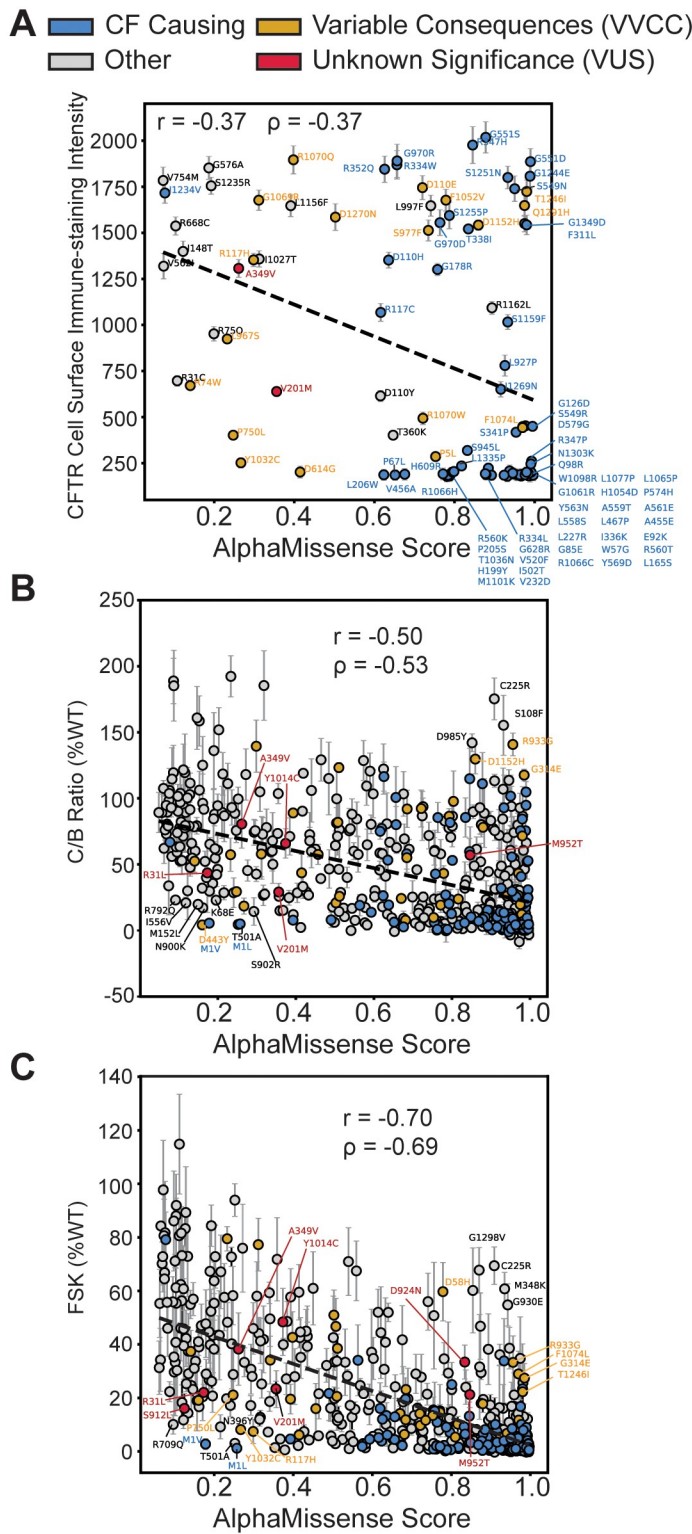

**Fig 3. Benchmarking AlphaMissense against CFTR *in vitro* functional metrics. A**. AM score plotted against deep mutational scanning data for 100 missense variants in HEK293T cells [23]. The y-axis represents the cell surface immune-staining intensity of CFTR and thus is indicative of the CFTR trafficking levels to the cell surface. A slight inverse linear correlation was observed (Pearson Coefficient r = -0.37, Spearman Coefficient ρ = -0.37). Some off-axis variants such as G551D are gating variants and thus fail to experience aberrant trafficking in the HEK293T cell

background but still exhibit impaired channel function. CF-causing variants are shown in blue, variants of variable clinical consequence (VVCC) are shown in yellow, and variants of unknown significance (VUS) are shown in red (as annotated in CFTR2). **B**. AM score plotted against CFTR western blot C:B band ratio in percent WT for 538 missense variants [33]. Several off-axis variants are highlighted. Color annotation as in **A**. Inverse linear correlation (Pearson Coefficient r = -0.50, Spearman Coefficient ρ = -0.53) improves for this larger dataset. Of note, VUS D924N and S912L were filtered out of this dataset due to an SEM >30, indicative of poor reproducibility across replicates (S4A and S4B Fig). **C**. AM score plotted against Forskolin induced CFTR current in percent WT for 546 missense variants [33]. Color annotation as in **A**. Inverse linear correlation (Pearson Coefficient r = -0.70, Spearman Coefficient ρ = -0.69) is higher than the function data compared to trafficking alone. All seven CFTR2 VUSs are highlighted.

## IV. AlphaMissense cannot predict CFTR variant theratype

Given the rapid and continuous emergence of novel CFTR variants detected by next-generation sequencing technologies, as well as a robust pipeline of new modulators and other CFTR-directed treatments under development, the need remains for optimized approaches to CF precision therapeutics. CFTR variant theratyping is an established method for quantifying *in vitro* CFTR sensitivity to pharmacologic agents, results of which are utilized to predict treatment responses for genotype-matched patients [6, 52]. CF treatment involves two corrector compounds, VX-661 and VX-445, that likely bind directly to two unique sites on CFTR [14], show distinct mechanisms, and hence distinct response profiles across variants. Thus, theratyping variant response remains an important task for CF personalized medicine.

We sought to determine whether AM offered any predictive power for CFTR theratyping, although this task is beyond the intended scope of AM. Theratype distinguishing plots were generated and colored by AM pathogenicity score to assess for potential patterns. We split VX-445-sensitive variants from VX-661-sensitive variants along a diagonal axis of best fit by plotting CFTR immunostaining intensity for VX-445 versus VX-661 (Fig 4A). Variants responsive to VX-445 fell above the dotted line, and variants responsive to VX-661 fell below the dotted line [23]. Variants were then colored by AM pathogenicity score, although the color distribution across the responsive spectrum revealed little discernable patterns (Fig 4A). We also plotted basal CFTR immune staining intensity versus VX-661, VX-445, or the combination thereof, then shaded variants by AM pathogenicity (S6A–S6C Fig). Similarly, AM scores showed little-to-no color patterns and appear randomly distributed.

Next, we used the theratyping study CFTR functional data [33] to plot the VX-445 + VX-661 FSK-mediated response (% of WT) versus basal activity, then colored the values by AM pathogenicity score (Fig 4B). Benign variants fell along a linear diagonal, suggesting that benign predicted variants all experience a linear response to CFTR correctors. We speculate this shift may reflect well-documented WT modulator response, implying an inherent stabilizing effect of VX-445 and VX-661. C:B band ratio response colored by AM score portrayed a random distribution of score color (S7A Fig). Pathogenic predicted variants in both plots show a random distribution. To determine whether theratype was predicted by variant structural location within CFTR, combined with AM score, we subdivided the plot in Fig 4B by domain (S7B–S7E Fig). Each domain individually showed a similar random distribution of score colors. Finally, we calculated relative degree CFTR correction by subtracting basal FSK (% of WT) from VX-445+VX-661 correction FSK (% of WT) and plotted this difference against AM score (Fig 4C). Again, no obvious pattern was observed. In summary, we found AM score afforded little predictive power for profiling pharmacologic responsiveness of CFTR variants. However, AM score could potentially be a useful machine learning feature for future theratype prediction methods.

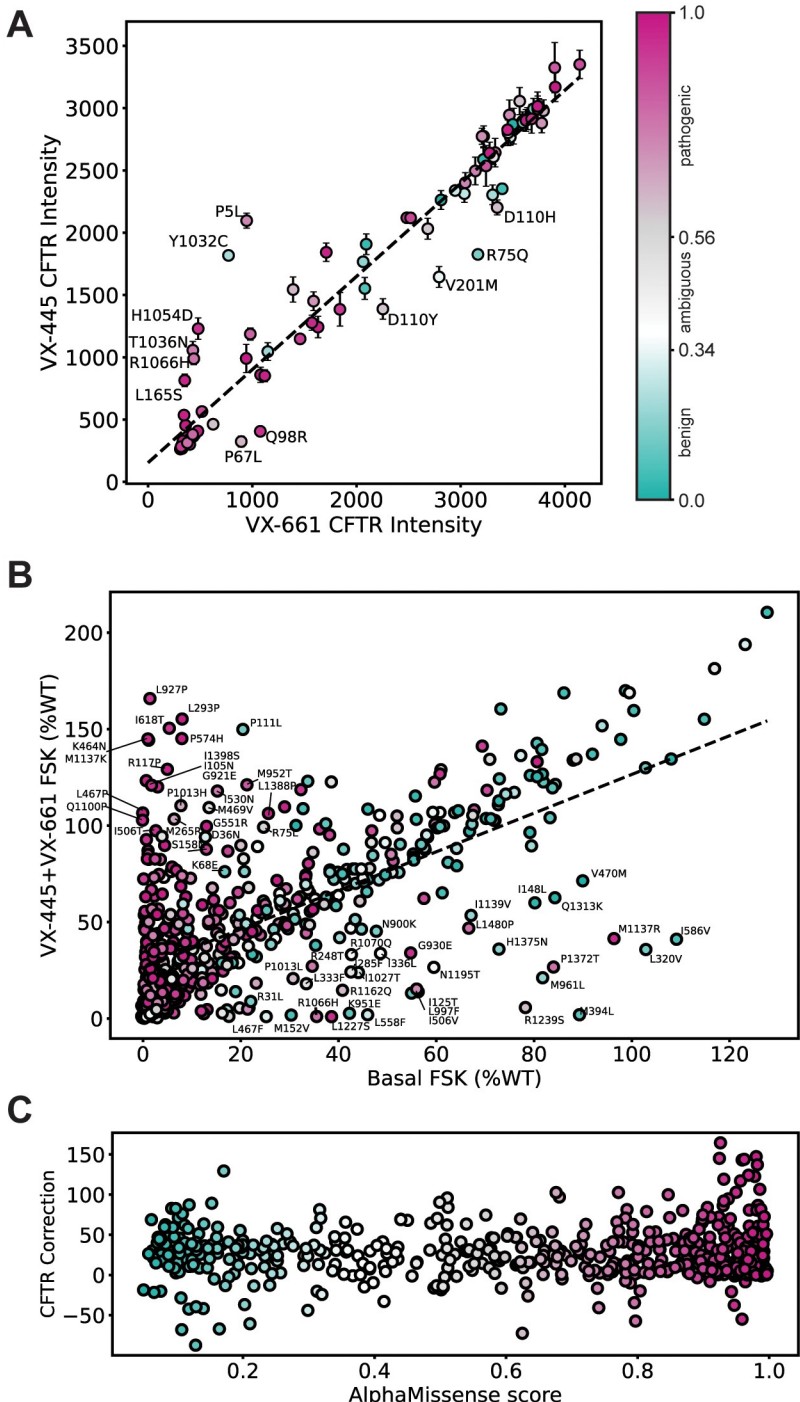

**Fig 4. CFTR theratype plots colored by AlphaMissense pathogenicity score. A**. CFTR cell surface immune staining intensity comparing treatment with VX-661 versus VX-445 correctors with the dotted line representing equivalent response to both correctors [23]. Variants that fell below the best-fit dotted line are selectively responsive to VX-661 while variants above the dotted line were selectively responsive to VX-445. The AlphaMissense pathogenicity predictions (color gradient) show no correlation with the corrector response patterns for CFTR variants. Variants with a score from 0.56–1.00 were classified by AM as pathogenic (violet), variants with a score from 0.34–0.56 were classified as ambiguous (grey), and variants with a score from 0.04–0.34 were classified as benign (green). Error bars represent the standard deviation of the cell surface immunostaining intensity. Selectively sensitive variants were annotated. **B**. Correlation of the basal CFTR activity versus CFTR activity when treated with dual corrector combination (VX-445, VX-661) from [33]. FSK-induced CFTR channel activity is expressed as % WT. Error bars were

excluded for clarity and AM predicted pathogenicity colored as in **A**. A line of benign variants emerged suggesting that AM benign predicted variants display a similar response to correctors that is dependent on their basal activity. Pathogenic variants, by contrast, showed little pattern or ability to predict theratype. Off-axis variants were annotated. **C**. Degree of functional CFTR correction was calculated by subtracting basal FSK (% of WT) levels from VX-445+VX-661 FSK (% of WT) levels and then plotted against AM score and colored by score. CFTR variants across the continuum of AM scores displayed variable functional responses to the corrector combination. AM predicted pathogenicity colored as in **A**.

## Conclusion

AlphaMissense has the exciting potential to aid with pathogenicity classification of rare and emerging variants identified during genetic screening. CF posited a valuable case study for evaluating AM performance because of abundance of clinical outcome data and *in vitro* variant classifications available. AM predicted pathogenicity of severe CF-causing variants well, albeit with a high false positive rate, and matched previous studies of CFTR variant pathogenicity in the NBD1/ICL4 interface [41]. However, AM performed modestly for pathogenicity predictions of VUSs and VVCCs, and the tool does not appear useful for CFTR theratype predictions. Again, for pathogenic missense variants, AM score correlated modestly with trafficking data and correlated well with channel activity functional data. Thus, predictions offer little information for distinguishing pathogenicity mechanism. AM may provide guidance in determining if polymorphisms in CFTR are benign, but performance on less severe disease variants indicate that caution must be taken when interpreting AM predictions. *In vitro* measurements on variant severity may aid in evaluating prediction quality and will remain necessary for CFTR theratyping.

## Methods

### Data curation and collection

AlphaMissense (AM) predictions for all single amino acid substitutions in the human proteome data was downloaded, gunzipped, and searched using vim text editor for CFTR accession number/Uniprot ID P13569. CFTR AM predictions were extracted into a separate file for analysis. ESM score predictions were downloaded from https://huggingface.co/spaces/ntranoslab/esm_variants by searching accession number P13569. EVE predictions were downloaded from https://evemodel.org/proteins/CFTR_HUMAN#variantsTableContainer by searching accession number P13569.

Cystic Fibrosis clinical outcome data was initially downloaded from the Variant List History tab on CFTR2.org. The table of 804 variants was filtered for 176 missense variants by removing in/dels, splicing variants, premature stop codons, etc. The patient information was manually curated by searching each variant and annotating the sweat chloride (mEq/L), pancreatic insufficiency rate (%), *P. aeruginosa* infection rate (%), lung function ages < 10 (FEV1%), lung function ages 10<20 (FEV1%), lung function ages >20 (FEV1%) (S1 Table). Lung function data proved too highly variable for comparison and was not used, but was still included in the Supporting Table for reference. CFTR2 definitions for these variants are as follows [19]: CF-causing: "A variant in one copy of the *CFTR* gene that always causes CF, as long as it is paired with another CF-causing variant in the other copy of the *CFTR* gene." Non-CF-causing: "A variant in one copy of the *CFTR* gene that does not cause CF, even when it is paired with a CF-causing variant in the other copy of the *CFTR* gene." Variant of Variable Clinical Consequence (VVCC): "A variant that may cause CF, when paired with a CF-causing variant in the other copy of the *CFTR* gene." Variant of Unknown Significance (VUS): "A

variant for which we do not have enough information to determine whether or not it falls into the other three categories."

*In vitro* modulator response data was downloaded from [33] and deep mutational scanning data downloaded from [23]. The 650 variants from [33] were filtered to 585 missense variants. ClinVar data was downloaded after searching for CFTR. Clinvar predictions were filtered for missense variants by removing, in/dels, stop codons, double missense variants, etc. Filtering yielded 1768 missense variant pathogenicity predictions (S2 Table). For performance comparison, the missense variants were filtered by clinical significance. We removed classifications such as *no interpretation*, *conflicting interpretations*, *uncertain significance*, and *drug response*. This left 219 variants classified as *pathogenic* or *likely benign* for performance evaluation and ROC plotting.

## Filtering experimental data

Experimental data from Bihler et al. [33] were filtered to exclude highly variable data based on the SEM due to lack of reproducibility. We plotted both the distribution of the data itself to look for outliers on the y axis of our correlation plots (Fig 3B and 3C) and the distribution of the SEM (S4 Fig). We labeled outliers with a C:B ratio of less than 30, but with a benign AM prediction of less than 0.3 (S4A Fig, Fig 3B). C:B band ratio SEM of greater than 30 were excluded from the analysis and not plotted for clarity, leaving 538 variants for analysis– 92% of the experimental data (S4B Fig). For the functional data, we labeled outliers of interest with a FSK % of WT of less than 30, but with a benign AM prediction of less than 0.3 (S4C Fig, Fig 3C). FSK % of WT SEM of greater than 20 were excluded from the analysis and not plotted for clarity, leaving 546 variants for analysis– 93% of the available experimental data.

## Analysis

Data were analyzed and plotted in Python 3. Raw excel files were imported and parsed using the Pandas data frame library and plots were generated with the matplotlib.pyplot and seaborn libraries. Pearson and Spearman correlation coefficients were calculated with the scipy.stats library using the pearsonr() and spearmanr() functions respectively. Plots were generated for all possible variants with available data for a given metric. The receiver operating characteristic (ROC) curve for AM pathogenicity predicted by AM was calculated against CFTR2 classification. CFTR2 classifies variants as CF causing, variable clinical consequence (VVCC), or non-CF causing. We equated the AM prediction pathogenic to CF causing, ambiguous to variable, and benign to non-CF causing. Since ROC is used for binary classification, all pairwise comparisons were considered. In each ROC curve, a different prediction (pathogenic, ambiguous, or benign) was taken to be a true positive, and the other two predictions to be false positives. Then the corresponding true positive and false positive rates were calculated by considering all possible score cutoffs for pathogenicity. Theratype discerning plots were generated to distinguish responsive from non-responsive variants graphically and colored by the variants respective AM score.

## Supporting information

**S1 Fig. AlphaMissense prediction of CFTR variants of unknown significance (VUS). A**. Conservation of residue in CFTR mapped on the structure (PDBID 5UAK) [25,26]. The abundance of green, representing low conservations scores in the TMDs stands in notable contrast the AM score predictions of pathogenicity in the TMDs. **B**. An overlay of the active state CFTR (PDB ID 6MSM) [30] and the AlphaFold prediction for CFTR [29] showing nearly perfect alignment of all resolved residues (1–409,435–637, 845–889, 900–1173, 1202–1451). We

calculated the root mean squared deviations (RMSD) of carbon backbone atoms between these two models in Chimera and found an RMSD of just 2.5 Å. **C**. Variants of unknown significance (VUS) displayed on CFTR structure (PDBID 5UAK) [25] demonstrates all unknown variants are in the transmembrane domains. Benign predicted mutations are shown in green, ambiguous predicted mutations in grey, and pathogenic predicted mutations are shown in purple. The two pathogenic predicted mutations both occur in transmembrane helix 8 (TH8) shown in orange. **D**. Receiver operating characteristic curve for AlphaMissense predictions of 115 Clinvar variants presented in the AlphaMissense benchmark. The average performance between pathogenic and benign variants is 95.8% as previously presented [21]. **E**. Receiver operating characteristic curve for AlphaMissense predictions of 209 variants downloaded directly from Clinvar [24], including 96 overlapping variants from the AlphaMissense benchmark. Again, the average performance is 95.8%. **F**. Receiver operating characteristic curve for AlphaMissense predictions of 113 variants downloaded directly from Clinvar [24] that did not overlap with variants from the AlphaMissense benchmark. Despite, not being trained on these ClinVar data, average performance is 95.8%. **G**. Overlap between AlphaMissense ClinVar benchmark set and our extended ClinVar set–showing 115 variants from AM and an additional 113 variants considered in **F**. Performance of AlphaMissense is very good across all permutations of ClinVar data considered. **H**. Due to the high number of VUS predictions in ClinVar[6] for CFTR missense mutations, we plotted the AlphaMissense score for all 1277 VUSs in ClinVar. We show 728 benign, 181 ambiguous, and 368 pathogenic variants as predicted by AM. Data is available in S2 Table.
(TIF)

**S2 Fig. Alternative prediction method performance and dataset overlap. A**. Receiver operating characteristic curve for ESM predictions [39] of 169 CFTR missense variants including 110 CF causing, 41 variable clinical consequence (VVCC), and 18 non-CF causing variants. For the pathogenic curve (violet), we considered a pathogenic prediction of a CF-causing variant a true positive. For the ambiguous curve (grey)—we considered the ambiguous prediction a VVCC a true positive. For the benign curve (bluegreen)–we considered the benign prediction of a non-CF causing variant as a true positive. **B**. Receiver operating characteristic curve calculated the same as in **A**. but using EVE missense variant predictions [40] of 169 CFTR missense variants colored as shown in **A**. **C**. Venn diagrams depicting the overlap of various datasets used throughout the study. We considered our expanded ClinVar dataset, the deep mutational scanning (DMS) dataset [23], our curated CFTR2 dataset, and the missense variants from the Bihler et al. dataset [33].
(TIF)

**S3 Fig. AlphaMissense prediction correlations with cystic fibrosis patient pathogenicity metrics by diagnosis. A**. AM score plotted against sweat chloride levels in milliequivalents per liter (mEq/L) for 85 missense variants classified as CF causing. The linear correlation (Pearson Coefficient r = 0.21, Spearman Coefficient ρ = 0.33) is reduced compared to the complete data set correlation shown in Fig 2A. **B**. AM score plotted against sweat chloride levels for 33 missense variants classified as variants of variable clinical consequence (VVCC). The linear correlation (Pearson Coefficient r = -0.12, Spearman Coefficient ρ = -0.11) is statistically insignificant. **C**. AM score plotted against pancreatic insufficiency rates in percent for 83 missense variants classified as CF causing. The correlation (Pearson Coefficient r = 0.32, Spearman Coefficient ρ = 0.44) was similar to the entire dataset in Fig 2B. **D**. AM score plotted against pancreatic insufficiency rates for 30 missense variants classified as VVCC. The correlation for these data (Pearson Coefficient r = -0.21, Spearman Coefficient ρ = -0.22) was statistically insignificant. **E**. AM score plotted against pseudomonas infection rates for 82 missense

variants classified as CF-causing. Linear correlation is reduced compared to the entire data set presented in Fig 2C (Pearson Coefficient r = 0.33, Spearman Coefficient ρ = 0.36). **F**. AM score plotted against pseudomonas infection rates in percent for 28 missense variants classified as VVCC. Linear correlation was insignificant (Pearson Coefficient r = 0.13, Spearman Coefficient ρ = 0.28).
(TIF)

**S4 Fig. Distributions of experimental data and error from Bihler et al. study for filtering purposes. A**. Histogram of the distribution of C-B band ratio of all 585 missense variants from the Bihler et al. study [33]. **B**. Histogram of the distribution of C-B band ratio SEM for all 585 missense variants from the Bihler et al. study. Variants with an SEM greater than 30 were excluded from analysis due to lack of experimental reproducibility and for plotting clarity. **C**. FSK %WT distribution plotted as a histogram for all 585 missense variants from the Bihler et al. study [33]. **D**. FSK %WT SEM distribution plotted as a histogram for all 585 missense variants from the Bihler et al. study. Variants with an SEM greater than 20 were excluded from analysis due to lack of experimental reproducibility and for plotting clarity.
(TIF)

**S5 Fig. AlphaMissense correlation with CFTR *in vitro* data from spatial covariance study. A**. Spatial covariance data from a previous study [46] for 62 missense variants plotted against AlphaMissense scores. Y axis represents the trafficking index as measured by a western blot trafficking assay when HEK293T cells were incubated at 37 ºC. A slight inverse linear correlation was observed (Pearson Coefficient r = -0.46, Spearman Coefficient ρ = -0.44). **B**. Spatial covariance data for 62 missense variants using the same trafficking index in **A**. except at 27 ºC, plotted against AlphaMissense scores. Again, an inverse linear correlation was observed (Pearson Coefficient r = -0.48, Spearman Coefficient ρ = -0.49) albeit slightly higher than at 37 ºC. **C**. AlphaMissense scores correlated with the spatial covariance data but using chloride conductance index described in [46], which measured channel activity at 37 ºC. We observed an increased correlation (Pearson Coefficient r = -0.58, Spearman Coefficient ρ = -0.54). **D**. AlphaMissense scores correlated with chloride conductance index at 27 ºC. We observed a slight correlation (Pearson Coefficient r = -0.50, Spearman Coefficient ρ = -0.53).
(TIF)

**S6 Fig. Deep mutational scanning data for VX-661 and VX-445 response colored by AlphaMissense pathogenicity score. A**. Basal CFTR surface immune staining versus VX-661 CFTR cell surface immune staining intensity [23]. Pathogenic variants score from 0.56–1.00 (violet), ambiguous variants score from 0.34–0.56 (grey), and benign variants score 0.04–0.34 (green). Error bars represent standard deviation. The distribution of pathogenicity colors throughout the plots suggested that AM pathogenicity prediction score failed to predict the VX-661 response. **B**. Basal CFTR surface immune staining versus VX-445 CFTR cell surface immune staining intensity. Colored the same as in **A**. Error bars represent standard deviation. Again, AM score failed to predict VX-445 response. **C**. Basal CFTR surface immune staining versus VX-661 + VX-445 CFTR cell surface immune staining intensity. Colored the same as in **A**. Error bars represent standard deviation. Finally, AM score failed to predict the combination of VX-661 and VX-445 response on a variant basis.
(TIF)

**S7 Fig. CFTR modulator response plots colored by AlphaMissense pathogenicity score reveals little predictive capabilities of AM in theratyping. A**. Basal mature CFTR (C band) to immature CFTR (B band) trafficking (C-B ratio) in percent WT versus modulator enhanced C/B ratio in percent WT from the Bihler et al. study [33]. Error bars were excluded for clarity.

Variants with an AlphaMissense pathogenicity prediction score from 0.56–1.00 were classified by AM as pathogenic (violet), a score from 0.34–0.56 as ambiguous (grey), and a score from 0.04–0.34 as benign (green). The distribution of colors across the plots indicated little predictive capability of AM on trafficking theratype. **B**. TMD1 variants only from Fig 4B. of the basal FSK CFTR activity in percent WT versus modulator enhanced FSK CFTR activity in percent WT from the Bihler et al. study [33]. Error bars were excluded for clarity and AM predicted pathogenicity colored as in **A**. **C**. NBD1 variants only from Fig 4B., data from the Bihler et al. study [33]. **D**. TMD2 variants only from Fig 4B., data from the Bihler et al. study [33]. **E**. NBD2 variants only from Fig 4B., data from the Bihler et al. study [33].
(TIF)

**S1 Table. CFTR2.org variant clinical outcome data.**
(XLSX)

**S2 Table. ClinVar CFTR variant data.**
(XLSX)

**S3 Table. In vitro CFTR trafficking and functional data from Bihler et al.**
(XLSX)

**S4 Table. Deep mutational scanning CFTR data.**
(XLSX)

## Author Contributions

**Conceptualization:** Eli Fritz McDonald, Lars Plate.

**Data curation:** Eli Fritz McDonald.

**Formal analysis:** Eli Fritz McDonald.

**Funding acquisition:** Eli Fritz McDonald, Kathryn E. Oliver, Jonathan P. Schlebach, Jens Meiler, Lars Plate.

**Investigation:** Eli Fritz McDonald.

**Methodology:** Eli Fritz McDonald.

**Project administration:** Lars Plate.

**Supervision:** Kathryn E. Oliver, Jonathan P. Schlebach, Jens Meiler, Lars Plate.

**Writing – original draft:** Eli Fritz McDonald.

**Writing – review & editing:** Eli Fritz McDonald, Kathryn E. Oliver, Jonathan P. Schlebach, Jens Meiler, Lars Plate.

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
