## [Decision Letter · Decision Letter 0]

30 Nov 2023

PONE-D-23-36353Benchmarking AlphaMissense Pathogenicity Predictions Against Cystic Fibrosis VariantsPLOS ONE

Dear Dr. Plate,

Thank you for submitting your manuscript to PLOS ONE. As you will note below, your manuscript was read and commented on by two experts in the field, both of whom--I am delighted to say--felt that your study was significant and would be of interest to the readers of the journal. However, there are some relatively minor concerns that, I agree, should be made to the paper in order to improve clarity. Most of these changes will be textual, but the inclusion of additional computational (e.g. comparisons to other predictive algorithms) and other information in Tables or Figures is needed. Nevertheless, this should not be too onerous...Therefore, we invite you to submit a revised version of the manuscript that addresses the points raised during the review process.

We look forward to receiving your revised manuscript, and should each of the comments be addressed, I am confident that I will be able to make an Editorial decision on the suitability of the manuscript for publication.

Thank you again for submitting your work to PLOS ONE.

Sincerely,

Jeffrey L Brodsky

Academic Editor

PLOS ONE

Journal Requirements:

 [This work was supported by R35 GM133552 (NIGMS), R01 HL167046 (NHBLI), R00HL151965 (NIH) and OLIVER22A0-KB (CFF). EFM was supported by a predoctoral fellowship F31 HL162483 (NHLBI) and Chemical-Biology Interface training grant T32 GM065086 (NIGMS).].  

[This work was supported by R35 GM133552 (NIGMS), R01 HL167046 (NHBLI), R00HL151965 (NIH) and OLIVER22A0-KB (CFF). EFM was supported by a predoctoral fellowship F31 HL162483 (NHLBI) and Chemical-Biology Interface training grant T32 GM065086 (NIGMS). ]

  [This work was supported by R35 GM133552 (NIGMS), R01 HL167046 (NHBLI), R00HL151965 (NIH) and OLIVER22A0-KB (CFF). EFM was supported by a predoctoral fellowship F31 HL162483 (NHLBI) and Chemical-Biology Interface training grant T32 GM065086 (NIGMS).].  

4. Please amend the manuscript submission data (via Edit Submission) to include author Jens Meiler.

5. We note that Figure 1A, S1a and S1b in your submission contain copyrighted images. All PLOS content is published under the Creative Commons Attribution License (CC BY 4.0), which means that the manuscript, images, and Supporting Information files will be freely available online, and any third party is permitted to access, download, copy, distribute, and use these materials in any way, even commercially, with proper attribution. For more information, see our copyright guidelines: http://journals.plos.org/plosone/s/licenses-and-copyright.

a. You may seek permission from the original copyright holder of Figure 1A, S1a and S1b to publish the content specifically under the CC BY 4.0 license. 

Reviewers' comments:

Reviewer's Responses to Questions

**Comments to the Author**

1. Is the manuscript technically sound, and do the data support the conclusions?

Reviewer #1: Yes

Reviewer #2: Yes

2. Has the statistical analysis been performed appropriately and rigorously? 

Reviewer #1: Yes

Reviewer #2: Yes

3. Have the authors made all data underlying the findings in their manuscript fully available?

Reviewer #1: Yes

Reviewer #2: Yes

4. Is the manuscript presented in an intelligible fashion and written in standard English?

Reviewer #1: Yes

Reviewer #2: Yes

5. Review Comments to the Author

Reviewer #1: In this study, McDonald et al used statistical methods and existing in vitro data to systematically evaluate the accuracy of the recently developed AlphaMissense technology in predicting the impact of missense variants within the cystic fibrosis transmembrane conductance regulator (CFTR) channel. The criteria the authors used for the assessment are: 1) Pathogenicity annotation from two databases, 2) Clinical features, 3) Protein trafficking and channel function, and 4) Theratypes based on responses to two CFTR correctors. Their data suggest that while the overall pathogenicity of CFTR variants can be predicted with high accuracy by AlphaMissense (AM), albeit slightly skewed towards a higher rate of false positives, AM is unable to distinguish between different mutation classes, theratypes, and clinical outcomes. Together, the study is thorough and the manuscript is well written, the methods used are sound, and the findings are useful for researchers and clinicians who would like to peruse AM to predict existing and new CFTR variants. Here are a few comments and suggestions to strengthen the manuscript and make it more accessible for readers outside the CFTR field:

1. It is unclear how much overlap of the AM training dataset (from ClinVar) with the list of variants in the CFTR2 repository and with the variants examined in the DMS and drug response studies. As this overlap would influence the accuracy of AM predictions, it would be helpful if the authors can provide a brief clarification on this.

Along a similar line of thought, how many CFTR ClinVar variants were included in the AM benchmark/ training set (Lines 95 and 98 suggest this might either be 115 or 104)? An ROC curve of all ClinVar variants (including the benchmark set) was performed in Supp Fig S1D, but a standalone ROC curve for only the variants not in the benchmark set might be useful.

2. Based on the results presented here, how much better (or worse) is AM in predicting CFTR variant pathogenicity compared to other methods such as ESM or EVE?

3. AM incorporates structural data from AlphaFold2 in its predictions. How reliable is this predicted structure compared to existing CFTR structures? Perhaps the poor correlation between AM scores and protein folding might be due to the quality of the predicted structure?

4. Other comments:

4.1. How are “variable consequence” variants defined in the CFTR2 repository? I think a brief clarification in the text would be helpful.

4.2. Table 1 lists all 7 variants from the CFTR2 repository that are of unknown consequence. Since the clinical features of individuals carrying these variants are further discussed in part II, it would be helpful to include these data in Supp Table S1 so that all data (such as # of alleles, frequency, sweat chloride levels, etc.) for all CFTR2 variants are available.

4.3. The authors brought up S912L as an example of a predicted benign variant, yet presenting with typical CF phenotypes. Has this variant been examined experimentally? Are individuals with this variant mostly heterozygous? Is it localized in a domain important for function and/or protein biogenesis? Similarly, were any variants from Table 1 examined in the in vitro studies or elsewhere?

4.4. If heterozygous data is readily available for other ClinVar/ CFTR2 variants, it would be useful to include this data in the supplemental tables, especially since the authors cited heterozygosity as a potential cause for the relatively poor performance of AM.

4.5. A couple sentences detailing the significance/ mode of action of VX-445 and VX-661 in part IV would be helpful for readers not in the immediate CFTR circle.

4.6. Are the correlation coefficients calculated in Figure 2 improved when only CF causing variants are taken into account?

4.7. Figure 2A has a grey box for “Other”, but no variants in this category were in this analysis.

4.8. Figure 2 legend, line 317: typo “variants were variable consequence were shown..”.

4.9. Line 194: What does the * refer to?

4.10. Are there titles and legends for the Supplemental Tables?

Reviewer #2: In this manuscript, McDonald et al explore the ability of the novel AlphaMissense technology to predict the pathogenicity of CFTR missense mutations. The study is quite relevant and, in general, well designed and presented. There are some aspects that should be considered before acceptance.

Major aspects

1. Descriptions of what was performed and is being presented as Results is in general very brief, demanding that the reader needs to constantly shift between the main text, the methods, and the figure legends. It would be very beneficial if the main text could include for each set of results one or two sentences, exposing what was done and is being presented.

2. When comparing data from CFTR2 and other sources, the authors analyze different number of CFTR variants – it is not clear why these specific sets of mutations were chosen (probably due to results being available). It would be good to add this rationale.

3. When analyzing the correlation of AM score with P.aeruginosa infection rate, it would be clearer to provide also separate plots for the different groups of individuals (CF-causing/Variable CC/Unknown) – probably as supplementary.

4. Results should be discussed considering work published by B Balch group – especially Anglès et al (2022) Comm Biol in which the authors present a spatial covariance analysis on the thermodynamic contribution of each residue to CFTR fold. Would it be feasible to add a comparative analysis of those results with the AM scores?

Minor aspects

5. When mentioning (l.112) the non-responsiveness of mutations at residue 560 to modulators, R560S can probably be added.

6. It is not clear what is meant by “CFTR fitness” (l.172). Do the authors mean “CFTR function”?

6. PLOS authors have the option to publish the peer review history of their article (what does this mean?). If published, this will include your full peer review and any attached files.

Reviewer #1: No

Reviewer #2: No

---

## [Author Response · Author response to Decision Letter 0]

3 Jan 2024

Reviewer #1

In this study, McDonald et al used statistical methods and existing in vitro data to systematically evaluate the accuracy of the recently developed AlphaMissense technology in predicting the impact of missense variants within the cystic fibrosis transmembrane conductance regulator (CFTR) channel. The criteria the authors used for the assessment are: 1) Pathogenicity annotation from two databases, 2) Clinical features, 3) Protein trafficking and channel function, and 4) Theratypes based on responses to two CFTR correctors. Their data suggest that while the overall pathogenicity of CFTR variants can be predicted with high accuracy by AlphaMissense (AM), albeit slightly skewed towards a higher rate of false positives, AM is unable to distinguish between different mutation classes, theratypes, and clinical outcomes. Together, the study is thorough and the manuscript is well written, the methods used are sound, and the findings are useful for researchers and clinicians who would like to peruse AM to predict existing and new CFTR variants. Here are a few comments and suggestions to strengthen the manuscript and make it more accessible for readers outside the CFTR field:

We thank reviewer 1 for their overall very positive feedback and recommendation.

Reviewer #1, Major Comments:

R1.1. It is unclear how much overlap of the AM training dataset (from ClinVar) with the list of variants in the CFTR2 repository and with the variants examined in the DMS and drug response studies. As this overlap would influence the accuracy of AM predictions, it would be helpful if the authors can provide a brief clarification on this. Along a similar line of thought, how many CFTR ClinVar variants were included in the AM benchmark/ training set (Lines 95 and 98 suggest this might either be 115 or 104)? An ROC curve of all ClinVar variants (including the benchmark set) was performed in Supp Fig S1D, but a standalone ROC curve for only the variants not in the benchmark set might be useful.

Our response: We thank Reviewer 1 for this helpful suggestion. Firstly, to clarify and quantify the overlap between the dataset, we now include a systematic set of Venn diagrams for the variants between all possible combinations of datasets including the expanded ClinVar dataset, CFTR2, DMS, and the study by Bihler et al. – leading to 6 total Venn diagrams. We included these Venn diagrams in Supplemental Figure 2C. 

Page 5, Line 229: “Variable, albeit high, overlap was observed between the CFTR2 dataset and the in vitro data sets discussed below (Supplemental Figure S2C).”

Secondly, upon closer inspection of the overlap between the AM ClinVar benchmark set and our expanded ClinVar set – we found an overlap of 96 variants, not 104 due to some repeated variants in the data set. We now also include a Venn diagram of the overlap between the AlphaMissense ClinVar dataset and our expanded dataset in Supplemental Figure S1G. This revealed 113 additional variants not included in the AM benchmark, we subsequently calculated the ROC curve for these variants revealing a pathogenic AUC of 1.00 and a benign AUC of 0.92. We now include this in Supplemental Figure S1F and added the following sentences to the manuscript to highlight these data.

Page 4, Line 145: “Finally, we plotted a ROC curve for 113 ClinVar variants not included in the AM benchmark set, which revealed a >90% accuracy and indicates AM performs well on ClinVar data outside of the training set (Supplemental Figure S1F-G).” 

R1.2. Based on the results presented here, how much better (or worse) is AM in predicting CFTR variant pathogenicity compared to other methods such as ESM or EVE?

Our response: We agree with Reviewer 1 that it is important to compare prediction methods. We downloaded the CFTR predictions for both the suggested methods and calculated the AUC ROC curve in the same manner as calculated for CFTR. Interestingly, this showed similar performance for pathogenicity predictions (ESM AUC = 0.76, EVE AUC = 0.73) but a slightly lower benign prediction capability (ESM AUC = 0.78, EVE AUC = 0.78). We added the following sentences to the manuscript to highlight these findings.

Page 4, Line 150-156: “AM performance was also compared to two other pathogenicity prediction tools, Evolutionary Scale Modeling (ESM)39 and Evolutionary model of Variant Effect (EVE)40 (Supplemental Figure S2A-B). ROC AUCs of benign variants, the AM value (0.91) was higher than those obtained for ESM (0.78) or EVE (0.78). A similar observation was made for pathogenic ROC AUCs, with AM (0.80) slightly above ESM (0.76) or EVE (0.73). ROC AUCs for ambiguous variants were nearly uniform across all methods (AM, 0.66; ESM, 0.65; EVE, 0.64). AM therefore offers a slight advantage for predicting pathogenic or benign variants and less utility regarding ambiguous variants. ”

R1.3. AM incorporates structural data from AlphaFold2 in its predictions. How reliable is this predicted structure compared to existing CFTR structures? Perhaps the poor correlation between AM scores and protein folding might be due to the quality of the predicted structure?

Our response: We agree with Reviewer 1 that a poor AlphaFold2 prediction could drive poor correlation, however we note that the AF2 prediction of CFTR is quite good. Specifically, AF2 prediction available on Uniprot has a root mean squared deviation of just 2.5 Å from the active state cryo-EM model (PDB ID 6MSM) for resolved residues (1-409,435-637, 845-889, 900-1173, 1202-1451). We added the following sentence to the manuscript to highlight this. 

Page 3, Line 123-125: “A high false positive rate may be attributed to a poor AlphaFold2 (AF2) predicted structure of CFTR. However, the AF2 predicted CFTR29 shows a root mean squared deviation of just 2.5 Å from the active state cryo-EM model (PDB ID 6MSM, resolution 3.2 Å30) (Supplemental Figure S1B).”

Reviewer #1, Minor Comments:

R1.4.1. How are “variable consequence” variants defined in the CFTR2 repository? I think a brief clarification in the text would be helpful.

Our response: We agree this is an important technical term to define, we added the following sentences to the first results to clarify “variable consequences”. As well as detailed definitions of all variants from CFTR2.org to the methods section. 

Page 3, Line 109: “VVCC are defined by CFTR2 as variants that may cause CF when found heterozygous with CF-causing variants, which results in variable clinical diagnosis of CF, e.g., a person with a VVCC and a CF-causing variant may or may not present with CF19.”

Page 8, Line 335: “CFTR2 definitions for these variants are as follows19: CF-causing: “A variant in one copy of the CFTR gene that always causes CF, as long as it is paired with another CF-causing variant in the other copy of the CFTR gene.” Non CF-causing: “A variant in one copy of the CFTR gene that does not cause CF, even when it is paired with a CF-causing variant in the other copy of the CFTR gene.” Variant of Variable Clinical Consequence (VVCC): “A variant that may cause CF, when paired with a CF-causing variant in the other copy of the CFTR gene.” Variant of Unknown Significance (VUS): “A variant for which we do not have enough information to determine whether or not it falls into the other three categories.””

R1.4.2. Table 1 lists all 7 variants from the CFTR2 repository that are of unknown consequence. Since the clinical features of individuals carrying these variants are further discussed in part II, it would be helpful to include these data in Supp Table S1 so that all data (such as # of alleles, frequency, sweat chloride levels, etc.) for all CFTR2 variants are available.

Our response: We thank Reviewer 1 for alerting us to the point. The variants of unknown consequence were not included in Supplemental Table S1 by mistake. We have now included these variants as well as all variants plotted in Figure 2, such that the Figure 2 plot can easily be reproduced by others from Table S1 with all available data on alleles and clinical outcomes. 

R1.4.3. The authors brought up S912L as an example of a predicted benign variant, yet presenting with typical CF phenotypes. Has this variant been examined experimentally? Are individuals with this variant mostly heterozygous? Is it localized in a domain important for function and/or protein biogenesis? Similarly, were any variants from Table 1 examined in the in vitro studies or elsewhere?

Our response: We thank Reviewer 1 for this helpful suggestion. Although S912L was not studied experimentally in the DMS study and was filtered out of the drug response study due to lack of reproducibility, a literature search revealed past experimental studies on this variant and others from Table 1. As there are only 7 patients with S912L in the CFTR2 database, we would venture a guess that they are all heterozygous, although this is not documented specifically. We show the location in the CFTR structure of all variants of unknown significance in Supplemental Figure S1C, although we did not draw attention to this in the first submission of the manuscript – we added a sentence below to highlight this. Additionally, we have added the paragraph below to the manuscript to highlight previous studies of all missense variants of unknown consequence from CFTR2. 

Page 3, Line 126: “The location of these variants in the CFTR structure is shown (Supplemental Figure S1C).”

Page 3, Line 127: “Benign predicted R31L disrupts the arginine framed tripeptide motif at R29-R31, important for folding evaluation prior to ER export31 and may affect endocytosis rates32. V201M was ambiguously predicted, consistent with our previous report describing this variant as mildly mis-trafficked and selectively sensitive to VX-66123. A439V (benign prediction) and Y1014C (ambiguous prediction) showed trafficking and function slightly below WT33 suggesting these variants are benign. Benign predicted variant S912L lies close to the CFTR glycosylation sites at N894 and N900, thus we speculated this mutation could interfere with glycan processing. Nevertheless, S912L trafficking and function remained sufficient compared to WT in vitro34. 

Variants D924N and M952T, both located in transmembrane helix 8, are predicted as pathogenic (Supplemental Figure S1C). D924N resides in the potentiator binding hotspot35,36 and, according to clinical data, may cause pancreatic insufficiency but not lung disease37. M952T displays robust functional expression in vitro38, and two patients with an M952T/F508del genotype exhibit normal chloride transport measured from intestinal mucosa38 – suggesting this variant is likely not pathogenic, despite the AM prediction.” 

R1.4.4. If heterozygous data is readily available for other ClinVar/ CFTR2 variants, it would be useful to include this data in the supplemental tables, especially since the authors cited heterozygosity as a potential cause for the relatively poor performance of AM.

Our response: Although this would be nice to present, heterozygous data is not available in CFTR2 or ClinVar. 

R1.4.5. A couple sentences detailing the significance/ mode of action of VX-445 and VX-661 in part IV would be helpful for readers not in the immediate CFTR circle.

Our response: We agree with Reviewer 1 that adding some details on the CFTR corrector compounds VX-445 and VX-661 will increase the accessibility of our manuscript to a broader readership. We added the following sentences to the introduction and a condensed summary to section IV of Results.

Page 2, Line 53: “At present, elexacaftor-tezacaftor-ivacaftor (ETI) is the best available highly effective modulator therapy for CF. This triple combination is clinically approved for ~170 CFTR variants, including the most commonly reported allele, deletion of phenylalanine 508 (F508del)8–11. ETI is composed of one gating potentiator (ivacaftor, VX-770) and two protein maturation correctors, tezacaftor (VX-661) and elexacaftor (VX-445). The corrector compounds have been suggested to directly bind unique subdomains of CFTR: VX-661 to TMD112,13, and VX-445 to the N-terminal lasso and TMD214,15. Correctors contribute intermolecular interactions that favor the properly folded, trafficking competent state of CFTR. Due to the distinct binding sites, VX-661 and VX-445 elicit different mechanisms of action and confer variable theratype responses. Thus, profiling CFTR variant theratypes to these and other emerging modulators remains an important priority for CF personalized medicine.” 

Page 6, Line 274: “CF treatment involves two corrector compounds, VX-661 and VX-445, that likely bind directly to two unique sites on CFTR14, show distinct mechanisms, and hence distinct response profiles across variants. Thus, theratyping variant response remains an important task for CF personalized medicine.”

R1.4.6. Are the correlation coefficients calculated in Figure 2 improved when only CF causing variants are taken into account?

Our response: We thank reviewer 1 for this suggest, however the correlation coefficients decrease by 0.05-0.2 when considering only the CF-causing variants. The exception is pancreatic insufficiency rates, which show a very slight increase in correlation about ~0.01-0.3 which we interpret to be inconsequential. 

 All data CF causing variants only

sweat chloride r = 0.46, ρ = 0.48 r = 0.21, ρ = 0.33

pancreatic insufficiency rates r = 0.31, ρ = 0.41 r = 0.32, ρ = 0.44

pseudomonas infection rates r = 0.38, ρ = 0.43 r = 0.33, ρ = 0.36

As also requested by Reviewer 2 comment 3, we included plots of sweat chloride, pancreatic insufficiency rates, and pseudomonas infection rates separated by CF-causing vs. variable consequence variants in Supplemental Figure S3. We conclude that correlation is best when all data is included and highlight this in the manuscript in the following sentences. 

Page 4, Line 193: “When considering CF-causing or VVCC separately, we note a reduced correlation between sweat chloride levels and AM scores (Supplemental Figure 3A-B), suggesting AM captures the trend across all variant types rather than performing better on pathogenic variants.”

Page 4, Line 200: “However, considering CF-causing and VVCCs separately failed to change the correlation for pancreatic insufficiency (Supplemental Figure 3C-D).”

Page 4, Line 205: “yet correlation was again reduced when only CF-causing or VVCCs were separately considered (Supplemental Figure 3E-F).”

R1.4.7. Figure 2A has a grey box for “Other”, but no variants in this category were in this analysis.

Our response: Indeed, there are no variants classified as “Other” by the CFTR2.org database and Figure 2 only contains CFTR2.org data. Thus, we removed this legend label. 

R1.4.8. Figure 2 legend, line 317: typo “variants were variable consequence were shown..”.

Our response: We corrected this typo to “variant of variable consequence were shown…”. 

R1.4.9. Line 194: What does the * refer to?

Our response: We apologize for this error, this asterisk indicated a citation needed to be inserted here, we added the proper citation.

R1.4.10. Are there titles and legends for the Supplemental Tables?

Our response: We added a list of Supplemental Tables to the main text. Additionally, we added a description tab to each excel file containing a brief legend/description of the data present in the table.

Page 18, Line 482: 

“List of Supplemental Tables 

Supplemental Table S1: CFTR2.org variant clinical outcome data

Supplemental Table S2: ClinVar CFTR variant data

Supplemental Table S3: In vitro CFTR trafficking and functional data from Bihler et al

Supplemental Table S4: Deep Mutational Scanning CFTR data”

 

Reviewer #2

In this manuscript, McDonald et al explore the ability of the novel AlphaMissense technology to predict the pathogenicity of CFTR missense mutations. The study is quite relevant and, in general, well designed and presented. There are some aspects that should be considered before acceptance.

Thank you to reviewer 2 for this positive assessment of our study.

Reviewer #2, Major Comments

R2.1. Descriptions of what was performed and is being presented as Results is in general very brief, demanding that the reader needs to constantly shift between the main text, the methods, and the figure legends. It would be very beneficial if the main text could include for each set of results one or two sentences, exposing what was done and is being presented.

Our response: We agree with Reviewer 2 that including a brief description of Methods in the Results section would help with readability. At the same time, we seek to maintain a succinct Results section for accessibility to a broad readership, including those on the clinical side of CF. To compromise we have included the following clarifying sentences in the Results to describe the ROC curve calculation, the CFTR2 database curation, and in vitro data filtering rationale. 

Page 3, Line 116: “Briefly, all pairwise comparisons were considered - pathogenic, ambiguous, or benign were taken in turn to be a true positive. The alternative two predictions for a specific comparison were taken to be false positives. We considered pathogenic to predict CF-causing, ambiguous to predict VVCC, and benign to predict non-CF causing, VUS were not used. While looping through all possible score thresholds, the corresponding true positive and false positive rates were calculated and plotted.”

Page 4, Line 184: “Briefly, CFTR2.org data were downloaded from the Variant List History tab and filtered for 176 missense variants (169 classified and 7 VUS). Then, clinical outcome data were manually assembled by searching each variant and recording the sweat chloride (mEq/L), pancreatic insufficiency rate (%), P. aeruginosa infection rate (%), and lung function (forced expiratory volume in one second (FEV1), % predicted).”

Page 5, Line 242: “We removed highly variable experimental data with a standard error of the mean (SEM) greater than 30. Most CFTR variants show a C:B ratio less than 30% of WT, indicating a lack of reproducibility for these measurements with higher variability (8% of data points removed, 92% retained).” 

Page 5, Line 251: “Again, highly variable experimental data were filtered out considering an SEM greater than 20 as most variants were less than 20% of WT (Supplemental Figure 4, See Methods), leaving 93% of the experimental data for comparison to AM.” 

R2.2. When comparing data from CFTR2 and other sources, the authors analyze different number of CFTR variants – it is not clear why these specific sets of mutations were chosen (probably due to results being available). It would be good to add this rationale.

Our response: We agree with Reviewer 2, that the number of variants compared between plots is somewhat convoluted. For clinical outcome data from CFTR2, this is simply a matter of data availability. Some variants with low patient measurements have insufficient data for one or more of the metrics considered, e.g. sweat chloride, pancreatic insufficiency, and pseudomonas infection rates. We added the following clarifications below shown in italics. 

Page 4, Line 182: “We curated the clinical outcomes for all CFTR missense variants with available data (Supplemental Table 1, See Methods).”

Page 4, Line 189: “First, we plotted AM score versus CF sweat chloride levels for 123 missense variants with sweat chloride values reported (Figure 2A).”

Page 5, Line 197: “Next, we plotted AM score versus pancreatic insufficiency rates for 116 missense variants present on at least one allele of persons with CF with CFTR2 outcomes reported (Figure 2B).”

Page 4, Line 170: “Finally, we plotted AM score versus P. aeruginosa infection rates for 114 missense variants on at least one allele with CFTR2 outcomes reported (Figure 2C).”

R2.3. When analyzing the correlation of AM score with P. aeruginosa infection rate, it would be clearer to provide also separate plots for the different groups of individuals (CF-causing/Variable CC/Unknown) – probably as supplementary.

Our response: We included correlation plots of all three clinical outcome metrics (sweat chloride levels, pancreatic insufficiency rates, and pseudomonas infection rates) separated by CF-causing and Variable consequence variants in Supplemental Figure S3, as also requested by Reviewer 1, comment 4.6. We draw attention to this Supplemental Figure in the context of pseudomonas infection rates in the following sentence. 

Page 5, Line 205: “yet correlation was again reduced when only CF-causing or VVCCs were separately considered (Supplemental Figure 3E-F).” 

R2.4. Results should be discussed considering work published by B Balch group – especially Anglès et al (2022) Comm Biol in which the authors present a spatial covariance analysis on the thermodynamic contribution of each residue to CFTR fold. Would it be feasible to add a comparative analysis of those results with the AM scores?

Our response: We agree with Reviewer 2 that the comparison of AM scores to the spatial covariance study from the Balch group represents an excellent dataset to include in our benchmark. In this study, Anglès et al (2022) Comm Biol, present a trafficking index and a chloride conductance index for 62 CFTR missense variants at 37 ºC and 27 ºC. We plotted the correlation of these in vitro metrics against the respective AlphaMissense score and included this as an additional Supplemental Figure 5. The following passage highlight this useful addition to the manuscript. 

Page 5, Line 257-265: “We verified the increased capability to predict CFTR function by correlating AM scores with a spatial covariance study (Supplemental Figure S5). This study describes trafficking (measured by western blot band shift assay) and chloride conductance indices and presents data for both metrics at 37 ºC and reduced temperature (27 ºC)46. Reduced temperature is a well-established method for partially rescuing F508del biogenesis51. We observed a modest correlation (Pearson coefficient: -0.46, Spearman Coefficient: -0.44) with trafficking index at 37 ºC, and a similar correlation at 27 ºC (Pearson coefficient: -0.48, Spearman Coefficient: -0.49) (Supplemental Figure S5A-B). Again, correlation increased when compared to chloride conductance index (Pearson coefficient: -0.58, Spearman Coefficient: -0.54 at 37 ºC vs. Pearson coefficient: -0.50, Spearman Coefficient: -0.53 at 27 ºC) (Supplemental Figure S5C-D).”

Reviewer #2, Minor Comments:

R2.5. When mentioning (l.112) the non-responsiveness of mutations at residue 560 to modulators, R560S can probably be added.

Our response: We thank Reviewer 2 for pointing this out, we added R560S and it now reads “R560T/K/S”.

R2.6. It is not clear what is meant by “CFTR fitness” (l.172). Do the authors mean “CFTR function”?

Our response: We agree with Reviewer 2 that the term “fitness” is generally ambiguous. We changed “fitness” to “function” or “functional” throughout the manuscript for clarity.

---

## [Editor Report · Decision Letter 1]

9 Jan 2024

Benchmarking AlphaMissense Pathogenicity Predictions Against Cystic Fibrosis Variants

PONE-D-23-36353R1

Dear Dr. Plate,

We’re pleased to inform you that your manuscript has been judged scientifically suitable for publication and will be formally accepted for publication once it meets all outstanding technical requirements.

Kind regards,

Jeffrey L Brodsky

Academic Editor

PLOS ONE
---

## [Editor Report · Acceptance letter]

17 Jan 2024

PONE-D-23-36353R1 

PLOS ONE

Dear Dr. Plate, 

I'm pleased to inform you that your manuscript has been deemed suitable for publication in PLOS ONE. Congratulations! Your manuscript is now being handed over to our production team.

Kind regards, 

on behalf of

Dr. Jeffrey L Brodsky 

Academic Editor

PLOS ONE